# *Coolia* Species (Dinophyceae) from the Tropical South Atlantic Region: Evidence of Harmfulness of *Coolia* cf. *canariensis* Phylogroup II

**Agatha Miralha** [1,2], **Silvia M. Nascimento** [1,3] and **Raquel A. F. Neves** [1,2,*]

1   Graduate Program in Neotropical Biodiversity, Institute of Biosciences, Federal University of the State
    of Rio de Janeiro (UNIRIO), Avenida Pasteur 458, Urca, Rio de Janeiro 22290-240, Brazil;
    htamiralha@edu.unirio.br (A.M.); silvia.nascimento@unirio.br (S.M.N.)
2   Research Group of Experimental and Applied Aquatic Ecology, Federal University of the State
    of Rio de Janeiro (UNIRIO), Avenida Pasteur 458-307, Urca, Rio de Janeiro 22290-240, Brazil
3   Laboratory of Marine Microalgae, Department of Ecology and Marine Resources, Institute of Biosciences,
    Federal University of the State of Rio de Janeiro (UNIRIO), Avenida Pasteur, Urca,
    Rio de Janeiro 22290-240, Brazil
*   Correspondence: raquel.neves@unirio.br; Tel.: +55-212244-5483

**Abstract:** Benthic dinoflagellates of the *Coolia* genus have been associated with cytotoxicity and lethal and sublethal effects on marine species. This study aimed to assess the harmful effects of *C.* cf. *canariensis* phylogroup II (PII) and *C. malayensis* strains through bioassays. Experimental exposures (24, 48, and 72 h) of *Artemia salina* nauplii to *Coolia* species (330–54,531 cells mL$^{-1}$) were performed independently. When a concentration-dependent response was achieved, additional experiments were carried out to evaluate the cell-free medium toxicity. The two *Coolia* species were harmful to *Artemia* nauplii, inducing significant mortality and sublethal responses. *Coolia* cf. *canariensis* PII was the most toxic species, inducing significant lethality at lower concentrations and shorter exposure times, followed by *C. malayensis*. Only the survival curves achieved after 24 and 48 h of exposure to *C.* cf. *canariensis* PII fitted to a concentration–response curve with valid LC$_{50}$s of 18,064 and 19,968 cells mL$^{-1}$, respectively. Moreover, extracellular compounds (i.e., culture filtrates) of *C.* cf. *canariensis* PII induced significant mortality to nauplii after 48 and 72 h. The toxicity of *C.* cf. *canariensis* PII was demonstrated for the first time using bioassays, and it was surprisingly higher than that of the *C. malayensis* strain, which was previously demonstrated to induce biological activity at the cellular and subcellular levels. Our findings highlight the harmful and lethal effects induced by *Coolia* cells and the importance of bioassays for toxicity assessments.

**Keywords:** Artemia test; bioassays; benthic dinoflagellate; harmful algae; lethality test; LC$_{50}$; phycotoxins



## 1. Introduction

Benthic dinoflagellates are microalgal cells that can support highly complex ecosystems, appearing attached to substrates such as macroalgae [1,2]. These groups of microalgae are globally distributed with greater species diversity in tropical and subtropical regions [3]. The genera *Ostreopsis* Schmidt (1902), *Prorocentrum* Ehrenberg (1834), *Coolia* Meunier (1919), *Gambierdiscus* Adachi & Fukuyo (1979), and *Amphidinium* Claraparède & Lachmann (1859) are the main representatives of the epi-benthic dinoflagellate assemblage and can co-occur at high densities in marine systems [2]. Some benthic dinoflagellates can synthesize toxic compounds, such as species from the genera *Gambierdiscus*, *Prorocentrum*, and *Ostreopsis* [4]. Some of these intracellular toxins are very potent and persistent in the food chain, causing harmful effects to marine life [4–7] and human health through the consumption of contaminated seafood [8].

The dinoflagellate genus *Coolia* is widely distributed in temperate and tropical regions [9]. Currently, this genus includes eight described species: *Coolia monotis* [10,11],

*C. tropicalis* [12], *C. areolate* [13], *C. canariensis* [14], *C. malayensis* [15], *C. santacroce* and *C. palmyrensis* [16], and *C. guanchica* [17]. *Coolia malayensis* has been found to be broadly distributed in temperate and tropical waters in the Atlantic, Pacific, and Indian oceans and the Mediterranean Sea, while *C. tropicalis, C. canariensis,* and *C. palmyrensis* are found to occur in the tropical areas of both the Atlantic and Pacific oceans [14,16,18–24]. *Coolia canariensis* is considered a species complex with cryptic diversity [16,18] and is phylogenetically divided into four clades [23]. While *C. canariensis* phylogroup III has a broad geographic distribution, strains from phylogroups I, II, and IV are currently known to occur only at one location [14,19,23], and phylogroup II is composed solely of one strain isolated from Trindade Island in Brazil [19].

*Coolia* species are not often related to the formation of high-biomass blooms, but a few blooms of *C. monotis* have been recorded in the Mediterranean Sea with maximum abundances of $1.5 \times 10^4$ cells $L^{-1}$ in the Egyptian coast and $2.5 \times 10^6$ cells $L^{-1}$ in the Gulf of Gabes [25]. In the oceanic Trindade Island (South Atlantic Ocean, Brazil), the abundance of the *Coolia* genus ranged from $1.0 \times 10^2$ cells $gFW^{-1}$ of the macroalgae *Dictyota mertensii* to $2.6 \times 10^3$ cells $gWW^{-1}$ of the macroalgae *Canistrocarpus cervicornis* [26]. Currently, five *Coolia* lineages are known to produce toxins: *C. canariensis* phylogroup IV, *C. tropicalis*, *C. malayensis*, *C. palmyrensis,* and *C. santacroce* [16,21,23,27–29]. Cooliatoxin, a yessotoxin analog, was the first toxin reported from a *Coolia tropicalis* strain [26]. Five compounds composed of less oxygen, compared to cooliatoxin and other analogs of yessotoxin were detected in *C. malayensis* from Okinawa in Japan [9]. Recently, a yessotoxin analog was detected in strains of *C. malayensis*, *C. canariensis* (phylogroup IV), and *C. palmyrensis* isolated from Guam, while other analogs were only found in *C. malayensis* and *C. canariensis* phylogroup IV [23]. The potent toxin 44-methylgambierone (i.e., maitotoxin-3 or MTX3), previously detected in some species of *Gambierdiscus* and *Fukuyoa*, was found in *C. malayensis* from Australia and New Zealand and *C. tropicalis* from Australia, Cook Islands, Brazil, Guam, and Hong Kong [21,30,31].

Hemolytic activity [31–34], cytotoxicity [16,34], hypothermia, and respiratory failure in mice [27,35] have been registered after exposure to extracts and/or lysates of *Coolia* species (e.g., *C. tropicalis*, *C. palmyrensis*, *C. santacroce,* and *C. malayensis*). Changes in the behavior of brine shrimps (nauplii and adults) and abnormal development in the pluteus larvae of sea urchins have been induced by *Coolia* exposure [21,23,36]. Exposure to *C. tropicalis* lysates has been shown to be lethal to medaka fish larva (*Oryzias melastigma*) by inducing hemolysis-associated toxicities and reducing fish heart rates [31]. Sublethal concentrations of *C. tropicalis* lysates have caused changes in the behavior and physiological performance of medaka larvae, as well as abnormalities in the early development of fish with changes in the expression of genes associated with apoptosis, inflammatory response, oxidative stress, and energy metabolism [37]. Moreover, crude extracts of a *C. malayensis* strain from Brazil (UNR-02) induced toxicity at the cellular and sub-cellular levels, leading to cell mass decrease and a significant depression in mitochondrial oxidative phosphorylation efficiency, which increased its susceptibility [38].

*Coolia* toxicity was previously considered species-specific [16], and intraspecific variability in toxin production is recognized in *C. malayensis* and *C. tropicalis* [2,23,39]. Considering interspecific toxicity, *C. malayensis* is generally reported as more toxic in bioassays than other species of the genus [23,36]. However, the induction of biological activity and toxicity to marine organisms is not directly related to the presence of detectable known intracellular toxins on *Coolia* isolates [23]. The present study aimed to assess the toxicity of two *Coolia* species (*C.* cf. *canariensis* phylogroup II (PII) and *C. malayensis*) isolated from tropical marine systems in the Brazilian coast and oceanic islands. Experimental exposures of *Artemia salina* nauplii to *Coolia* species at environmentally relevant concentrations (i.e., found in natural environments) were performed using laboratory bioassays. When the exposure to *Coolia* cells induced a concentration-dependent effect on nauplii survival, additional assays were performed to evaluate the toxicity of the cell-free medium (i.e., filtrates) of cultured strains to the brine shrimps. This study has direct implications for the United Nations Sustainable

Development targets (Goal 14—Life Below Water) by increasing the research efforts in marine sciences.

## 2. Materials and Methods

### 2.1. Dinoflagellates Cultures

Clonal cultures of *C.* cf. *canariensis* phylogroup II strain UNR-25 (GenBank MK109023) were isolated from Trindade Island, Brazil (20°29′22.2″ S, 29°20′04.2″ W), while *C. malayensis* strain UNR-02 (GenBank MK109022) was isolated from Armação dos Búzios, Rio de Janeiro state, Brazil (22°45′18″ S, 41°54′07″ W). These strains were maintained at the Marine Microalgae Culture Collection from the Federal University of the State of Rio de Janeiro (UNIRIO); detailed information concerning sampling and cells isolation is described in [19].

*Coolia* cultures were maintained at the exponential growth phase in filtered (glass-fiber filter, Millipore AP-40, Millipore Brazil, São Paulo, Brazil) seawater (FSW) with salinity of 34 and supplemented with L1 enrichment medium [40]. All stock cultures were kept in a temperature-controlled cabinet at 24 ± 2 °C, with a 12:12 h dark–light cycle and photon flux density of 60 µmol m$^{-2}$s$^{-1}$ provided by cool-white fluorescent tubes. Photosynthetically active radiation was measured with a QSL-100 quantum sensor (Biospherical Instruments, San Diego, CA, USA). Cell abundance was determined at the beginning of the experiments. Samples were preserved with neutral Lugol's iodine solution for cell counts (*n* = 3) using a Sedgewick-rafter chamber and observation in an inverted light microscope (Primovert, Zeiss, Göttingen, Germany).

### 2.2. Experimental Design

The maximum concentrations of *Coolia* cells detected in environmental samples from Brazil [26] and the Mediterranean Sea [25] were considered in the determination of the abundance range used in the bioassays with environmentally relevant concentrations of dinoflagellate cells. Just before the incubations, eight different cellular concentrations of each *Coolia* species were established by successive dilutions (factor of 2) of stock cultures in FSW. When necessary, cultures were concentrated by filtration in a polyamide mesh (mesh size = 10 µm) and resuspended in smaller volumes of FSW to reach a higher cellular concentration. Considering the maximum concentrations achieved in cultures, the ranges of cellular concentration were 330 to 42,270 cells mL$^{-1}$ for *C.* cf. *canariensis* PII and 926 to 54,531 cells mL$^{-1}$ for *C. malayensis.*

#### 2.2.1. Test Organism

Dried *Artemia salina* cysts (approximately 1 g) (Maramar, Brazil) were hatched in FSW (filtered in glass-fiber filter, Millipore AP-40, Millipore Brazil) with salinity of 34. FSW was aerated and maintained at 25 ± 1 °C under a 12:12 light–dark cycle. Newly hatched larvae (instar stages II and III) were sorted using a Pasteur pipette after 72 h of incubation for the bioassays. All the procedures applied in *Artemia* bioassays considered previous studies [41], using nauplii hatched from commercial cysts from the same lot and geographical origin, controlled abiotic conditions, and a standard test protocol for all the assays to reduce variability.

#### 2.2.2. Toxicity Tests

Bioassays were performed in 6-well plates containing ten *A. salina* nauplii in 10 mL of FSW (negative control) or FSW with *Coolia* cells (i.e., test solutions) by well [5]. Independent assays (true replicates) were performed for each species of *Coolia* (*C.* cf. *canariensis* and *C. malayensis*). In total, twelve control replicates (nauplii with only FSW) and four independent replicates by concentration were performed in each treatment (i.e., *Coolia* species).

Additionally, cell-free medium assays were performed to detect effects of extracellular compounds only when exposure to *Coolia* cells induced a concentration-dependent effect on nauplii survival. Cell-free medium was obtained by filtration using a glass-fiber filter (Millipore AP-40, Millipore Brazil) of *Coolia* cf. *canariensis* PII culture at the exponential growth phase with 126,000 total cells. Cell-free medium incubations were performed in

6-well plates immediately after filtration, using 10 mL of filtrate solution or FSW supplemented with L1-enriched medium (control) and ten nauplii by well. Cell-free medium treatment was performed in six independent replicates, and the control was performed using four replicates.

Nauplii survival (i.e., the number of alive individuals) was monitored after 24, 48, and 72 h of incubation with *Coolia* cells using a stereoscope microscope (Leica EZ4 HD). Additionally, changes in nauplii behavior (e.g., immobility, agitation, and swimming) were described during the incubations. Experiments were performed in a temperature-controlled cabinet at $24 \pm 2$ °C, with a 12:12 h light–dark cycle and photon flux density of $60$ µmol m$^{-2}$s$^{-1}$ provided by cool-white fluorescent tubes.

### 2.3. Data Analysis

The survival proportion (i.e., the number of alive individuals at $t_x$/individuals at $t_0$) was calculated by independent replicates at each exposure time. The arithmetic mean of independent replicates by the concentration of each *Coolia* species and negative controls were used to calculate the cumulative percentage of mortality. Data transformation was applied before statistical analysis using arcsine of the square root (proportion data) to conform the parametric test assumptions. A one-way analysis of variance (ANOVA) was performed to evaluate the influence of different concentrations of each *Coolia* species on the survival proportion of *A. salina* nauplii by exposure time (i.e., 24, 48, and 72 h), and Tukey's test was applied a posteriori. T-tests for independent samples were applied to compare the survival proportion of nauplii between the control and cell-free medium treatment after 24, 48, and 72 h of incubation. Kolmogorov–Smirnov and Levene tests were applied to assess the normality and homogeneity of variance, respectively, of data distribution. When the assumptions of the parametric test were not met, the non-parametric Kruskal–Wallis test was performed, followed by multiple comparisons. Parametric and non-parametric tests were considered statistically significant when $p \leq 0.05$. Statistical analyses were performed using the software Statistica 10 (StatSoft).

The LC$_{50}$ and 95% of confidence intervals (CIs) were determined using the survival data of replicates corrected by the mean of control data. A four-parameter logistic equation (variable slope) with the least-squares fitting method was applied after the log-transformation of the *x*-axis values (*Coolia* concentrations). The non-linear regression was carried out using the GraphPad Prism 5.01 software. LC$_{50}$ results were validated if the fitted concentration–response curves had an $R^2 \geq 0.80$ and if the percent fitting error of the LC$_{50}$ (FE, in percentage) was <40%. The % FE was calculated by the equation [42]:

$$\% \text{ FE} = \text{SE Log LC}_{50} \times \text{Ln10} \times 100$$

where SE = standard error.

## 3. Results

### 3.1. Nauplii Exposure to C. cf. canariensis PII

The exposure to *C.* cf. *canariensis* cells induced a significant effect on *A. salina* nauplii survival after 24 h (ANOVA, $F_{(8,44)} = 11.98$, $p \leq 0.001$), 48 h (ANOVA, $F_{(8,44)} = 33.17$, $p \leq 0.001$), and 72 h (Kruskal–Wallis, $H_{(8,44)} = 33.57$, $p \leq 0.001$) of exposure. Even exposure to the lower concentration of *C.* cf. *canariensis* tested (330 cells mL$^{-1}$) was enough to induce significant nauplii mortality after 24 h of exposure (Table 1). Moreover, all tested concentrations of the benthic dinoflagellate *C.* cf. *canariensis* PII induced sublethal effects on *A. salina* nauplii, such as abnormal swimming activity and mobility impairment. The movement of *A. salina* exposed to the higher dinoflagellate concentration (10,565 cells mL$^{-1}$) was extremely reduced, and nauplii showed slow appendage beats and the absence of displacement in the water column, possibly related to mucilage secreted by dinoflagellate cells at high concentrations.

**Table 1.** Cumulative mortality (%) of *Artemia salina* nauplii exposed to *Coolia* cf. *canariensis* PII after 24, 48, and 72 h. Data comprise the mean values of ten *A. salina* individuals from each independent replicate of control and eight dinoflagellate concentrations.

| *C.* cf. *canariensis* PII Concentration (Cells mL$^{-1}$) | Cumulative Mortality (%) | | |
|---|---|---|---|
| | **24 h** | **48 h** | **72 h** |
| 0 | 1.67 | 4.17 | 5.83 |
| 330 | 12.5 * | 35.0 * | 75.0 |
| 660 | 15.0 * | 40.0 * | 77.5 |
| 1320 | 12.5 | 50.0 * | 87.5 |
| 2641 | 5.0 | 37.5 * | 87.5 |
| 5283 | 5.0 | 45.0 * | 80.0 |
| 10,567 | 12.5 | 45.0 * | 95.0 * |
| 21,135 | 35.0 * | 75.0 * | 95.0 * |
| 42,270 | 50.0 * | 97.5 * | 100 * |

* Cumulative mortality significantly higher than control (Tukey's test or multiple comparisons, $p \leq 0.05$) at the same exposure time.

The survival data of *A. salina* nauplii showed a non-linear concentration-dependent response after 24 h and 48 h of exposure to cells of *C.* cf. *canariensis* PII; however, no clear concentration–response curve was observed after 72 h of exposure (Figure 1). Considering the adjustment to the four-parameter logistic model, LC$_{50}$ values were only determined for 24 h and 48 h of exposure. The LC$_{50}$ values were validated according to the previously described criteria, the fitting error (FE) of LC$_{50s}$ did not exceed 25.7%, and fitted dose–response curves did not have an $R^2$ lower than 0.829 (Table 2).

**Table 2.** Validated results of concentration–response LC$_{50}$ and 95% of confidence interval (CI) (cells mL$^{-1}$), with R$^2$ and fitting error (FE, %) values, after *Artemia salina* nauplii exposure to cells of *Coolia* cf. *canariensis* PII.

| Exposure Time (h) | LC$_{50}$ (Cells mL$^{-1}$) | 95% CI (Cells mL$^{-1}$) | $R^2$ | FE (%) |
|---|---|---|---|---|
| 24 | 18,064 | 10,815–30,171 | 0.853 | 23.3 |
| 48 | 19,968 | 11,343–35,150 | 0.829 | 25.70 |

Since the exposure to the cells of *C.* cf. *canariensis* PII induced a significant concentration-dependent effect on nauplii survival, additional assays were performed to test the toxicity of BEC to nauplii. Exposure to cell-free medium, after the filtration of *C.* cf. *canariensis* PII culture with 126,000 total cells, induced a significant lethal effect on nauplii after incubation for 48 h (*t*-test, *t*-value = 7.239, df = 8, *p*< 0.0001) and 72 h (*t*-test, *t*-value = 9.705, df = 8, *p* < 0.0001) (Figure 2). No significant difference between the control group (only FSW) and cell-free medium treatment was detected after 24 h of incubation (*t*-test, t-value = 1.699, df = 8, *p* = 0.128) (Figure 2).

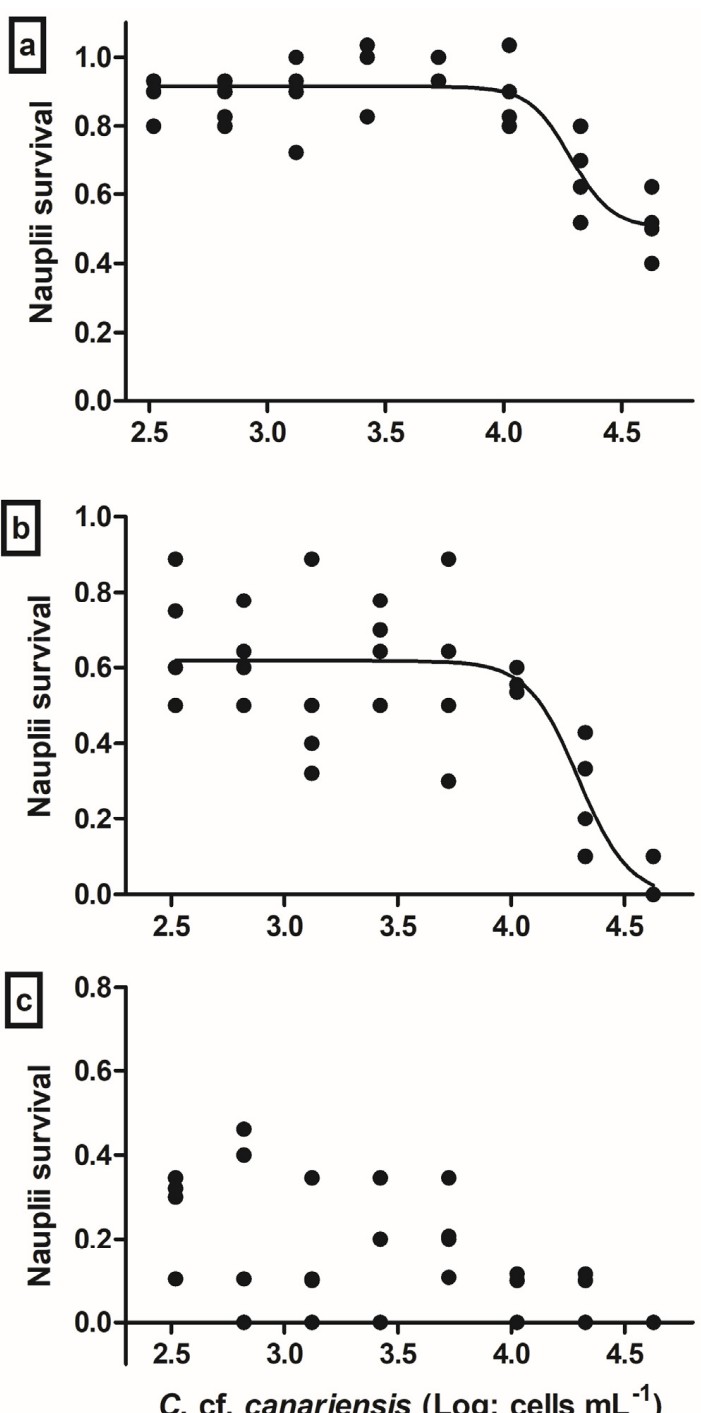

**Figure 1.** Survival (in proportion of negative controls) of *Artemia salina* nauplii exposed to *Coolia* cf. *canariensis* PII (Log, cells mL$^{-1}$) after 24 h (**a**), 48 h (**b**), and 72 h (**c**) of incubation. Data are shown as the values of independent replicates by concentration (*n* = 4), in black circles (●), and fit curve of validated assays (**a**,**b**) in black solid line. Since survival data after 72 h of exposure did not adjust to the four-parameter logistic model, there was no fit curve to be included in the graph (**c**). LC$_{50}$ and 95% of confidence intervals are shown in Table 2.

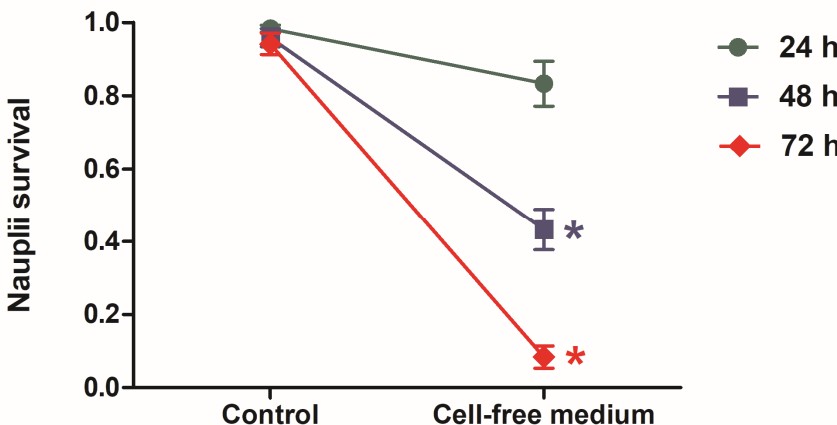

**Figure 2.** Mean ± SD values of survival (in proportion of alive individuals) after incubation of *Artemia salina* nauplii with FSW (control) and cell-free medium (extracellular compounds) for 24 h (●), 48 h (■), and 72 h (♦). * Corresponds to treatments significantly different from the control (*t*-test, $p < 0.0001$). Cell-free medium was obtained by the filtration of *C.* cf. *canariensis* PII culture with 126,000 total cells.

### 3.2. Nauplii Exposure to C. malayensis

Exposure to *C. malayensis* induced a significant effect on *A. salina* nauplii survival after 24 h (Kruskal–Wallis, $H_{(8,44)} = 29.012$, $p = 0003$), 48 h (Kruskal–Wallis, $H_{(8,44)} = 35.686$, $p < 0.0001$), and 72 h (Kruskal–Wallis, $H_{(8,44)} = 33.936$, $p < 0.0001$). Nauplii mortality was mostly detected at intermediary concentrations of *C. malayensis* (e.g., 1704–13,633 cells mL$^{-1}$) after 48 h and 72 h of exposure (Table 3). Moreover, exposure to cells of *C. malayensis* induced sublethal effects on *A. salina* nauplii, which showed abnormal swimming activity and mobility impairment, mostly at higher dinoflagellate concentrations, possibly related to mucilage secreted by dinoflagellate cells at high concentrations.

**Table 3.** Cumulative mortality (%) of *Artemia salina* nauplii exposed to *Coolia malayensis* after 24, 48, and 72 h. Data comprise the mean values of ten *A. salina* individuals from each independent replicate of control and eight dinoflagellate concentrations.

| *C. malayensis* Concentration (Cells mL$^{-1}$) | Cumulative Mortality (%) | | |
|---|---|---|---|
| | 24 h | 48 h | 72 h |
| 0 | 0 | 1.6 | 3.3 |
| 426 | 0 | 42.5 | 82.5 |
| 852 | 7.5 | 50.0 | 92.5 * |
| 1704 | 12.5 | 72.5 * | 95.0 * |
| 3408 | 2.5 | 52.5 | 87.5 |
| 6816 | 15.0 | 67.5 * | 97.5 * |
| 13,633 | 12.5 | 67.5 * | 92.5 * |
| 27,266 | 0 | 45.0 | 90.0 |
| 54,531 | 0 | 10.0 | 30.0 |

* Treatments in which mortality was significantly higher than in the control (multiple comparisons, $p \leq 0.05$) at the same exposure time.

No marked concentration–response curve was observed for the survival data of *A. salina* nauplii exposed to *C. malayensis* for the concentrations tested (426–54,531 cells mL$^{-1}$); thus, it was not possible to determine LC$_{50}$ values for *C. malayensis*. Independently of exposure time, a decrease in the tendency of nauplii to survive was shown at lower concentrations (426 and 1704 cells mL$^{-1}$), followed by a stabilization or increase in survival

response at intermediary and higher concentrations (Figure 3). Exposure to the benthic dinoflagellate *C. malayensis* induced changes in *A. salina* nauplii behavior, which showed abnormal swimming activity, mobility impairment, and agitation. Most of the effects were noticed at higher concentrations associated with reductions in nauplii mobility, possibly related to mucilage secreted by dinoflagellate cells at high concentrations.

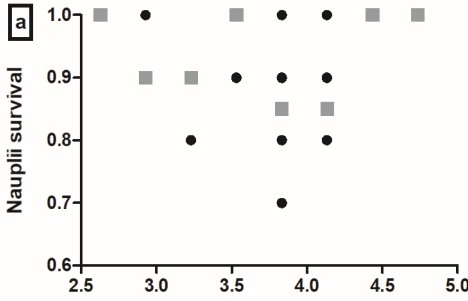

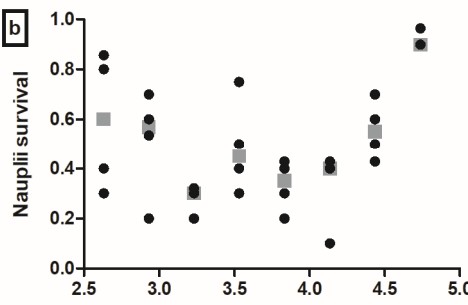

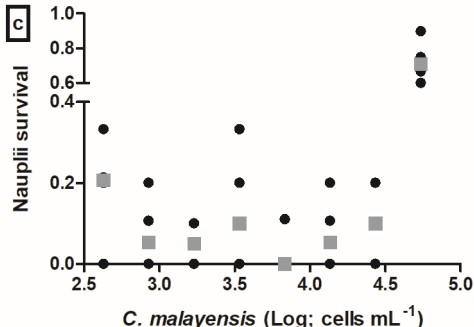

**Figure 3.** Survival (in proportion of negative controls) of *Artemia salina* nauplii exposed to *Coolia malayensis* (Log, cells mL$^{-1}$) after 24 h (**a**), 48 h (**b**), and 72 h (**c**) of incubation. Data are shown as values of independent replicates by concentration (*n* = 4), in black circles (●), and mean values by concentration (in grey squares, ■).

## 4. Discussion

In the present study, significant toxicity was induced by the strains of *Coolia* cf. *canariensis* PII and *C. malayensis*. Exposure to cells of *C.* cf. *canariensis* PII induced significant lethality to *A. salina* nauplii, even at the lower tested concentration (330 cells mL$^{-1}$) and shorter exposure time (24 h). Moreover, exposure to cells of this strain also induced sublethal effects on nauplii, and extracellular compounds (i.e., cell-free medium) induced lethal effects on nauplii after 48 and 72 h of exposure. *Coolia* cf. *canariensis* PII is composed solely of the strain tested in the current study, isolated from the oceanic Trindade Island in Brazil [19]. Little information is available in the literature on the biological activity of strains from other phylogroups of the *C. canariensis* species complex. Yessotoxin and its analogs were not detected in three strains from phylogroup I and III isolated from the Canary Islands (Spain), and the extracts of one strain, which was injected intraperitoneally

into mice, did not induce toxicity [14]. Bioassays performed with extracts of strains of *C.* cf. *canariensis* PIII from Hong Kong did not show significant lethality to nauplii of *Artemia franciscana* [36]. Moreover, direct exposure to cells of strains of *C.* cf. *canariensis* PIII from the Bay of Biscay (Spain) did not induce lethality to pluteus larvae of the sea urchin *Heliocidaris crassispina* [29]. Recently, yessotoxin analogs were detected in strains of *C.* cf. *canariensis* phylogroup IV isolated from Guam; however, their extracts did not induce biological activity and toxicity in *Artemia* bioassays [23]. Genetic differences separate strains from the *C.* cf. *canariensis* species complex in four phylogroups [23] which may also present differences in their biological activity and toxicity.

Surprisingly, the strain of *C.* cf. *canariensis* PII was more toxic than the strain of *C. malayensis* tested in the present study, considering both the exposure time and the minimum cellular concentration necessary to induce nauplius lethality. The lowest concentration of *C. malayensis* that induced a significant lethal effect was 852 cells mL$^{-1}$ after 72 h of exposure, while 48 h of exposure was the minimum time that induced significant mortality at 1704 cells mL$^{-1}$. An increase in the survival response of nauplii was detected at higher *C. malayensis* concentrations (independently of exposure time), which is possibly a hormetic effect (i.e., an overcompensation response to a disruption in homeostasis) [43]. Hormesis is a highly frequent phenomenon independently of a tested stressor, biological endpoint, and experimental model system [44]. It is not expected that a determined intensity of a specific stressor (e.g., a toxicant concentration) could induce similar hormetic responses in different biological systems; however, a hormetic effect was previously detected in *A. salina* individuals exposed to the chemical compound bisphenol A [45]. In contrast to our findings, *C. malayensis* is usually more toxic than other *Coolia* species [23,36], although [21] found a significant higher mortality rate of *A. salina* adults exposed to a strain of *C. tropicalis* that synthesizes gambierone (MTX3) at the maximum algal biomass tested (16,000 ng C mL$^{-1}$) compared to *C. malayensis* (19,300 ng C mL$^{-1}$), *C. santacroce* (15,000 ng C mL$^{-1}$), and *C. palmyrensis* (11,500 ng C mL$^{-1}$) after 72 h of incubations. In a previous study, the crude extract of the *C. malayensis* strain UNR-02 (i.e., the same used in the current study) induced a significant cell mass decrease in HepG2 and H9c2(2-1) cell lines at equivalent dinoflagellate concentrations of 5000–10,000 and 313–10,000 cells mL$^{-1}$, respectively, after 72 h of exposure [38]. Additionally, at the subcellular level, *C. malayensis* crude extract induced changes to mitochondrial membrane potential generation and fluctuations associated with the induction of mitochondrial permeability transition [38], reinforcing the concept that the extract of this strain is toxic at the subcellular level and in cellular-based assays.

In the present study, only the exposure to *C.* cf. *canariensis* PII induced a concentration-dependent effect on nauplii survival with validated LC$_{50}$ values of 18,064 and 19,968 cells mL$^{-1}$ for 24 h and 48 h of exposure, respectively. Since exposure to *Coolia malayensis* did not fit to a concentration–response curve, it was not possible to determine LC$_{50}$ values. There is no information concerning LC$_{50}$ results obtained in bioassays using the direct exposure of marine organisms to *Coolia* spp. cells as frequently occurs in natural marine environments, where species from this dinoflagellate genus are widely distributed in tropical and temperate regions [9]. The results obtained from the direct exposure to *Coolia* cells, applied in the current study, are not directly comparable to LC$_{50}$ results obtained in previous studies that performed the bioassays with *Coolia* lysates [36] and extracts [23]. A milder effect is expected to occur in individuals exposed to live dinoflagellate cells, without direct contact with intracellular toxins. *C. malayensis* lysates from strains S020 and Ve051 isolated from Hong Kong induced mortality significantly different from the control in larvae of brine shrimp and sea urchins, at concentrations up to 0.05 mg mL$^{-1}$, while lysates of *C. tropicalis* (strain S002) induced mortality significantly different from the control of sea urchin larvae at 0.075 mg mL$^{-1}$ [36]. The tested concentrations of lysates from *C.* cf. *canariensis* phylogroup III (strains W039 and Ve011) and *C. palmyrensis* (strains S017 and W085) did not significantly affect the survival of sea urchin larvae compared to the control [36]. The LC$_{50,48h}$ values of algal lysates of *Coolia* species was estimated for *A. franciscana* nauplii

incubated with two strains of *C. malayensis* (0.086–0.117 mg mL$^{-1}$), as well as for *H. cras-sispina* pluteus larvae incubated with two strains of *C. malayensis* (0.016–0.046 mg mL$^{-1}$), *C. tropicalis* (0.029–0.038 mg mL$^{-1}$), *C. palmyrensis* (0.023–0.049 mg mL$^{-1}$), and *C.* cf. *canarien-sis* phylogroup III (0.064–0.082 mg mL$^{-1}$) [36]. The toxicity of water-soluble extracts of *Coolia* species was tested in *Artemia* bioassays, and lethal and sublethal effects were detected in nauplii exposed to extracts of *C. malayensis*, while *C. palmyrensis* and *C.* cf. *canariensis* phylogroup IV did not exhibit a noxious effect [23]. A hydrophilic algal extract of a *C. tropicalis* strain, which produces 44-methylgambierone, induced a concentration effect on medaka fish larvae survival with an LC$_{50,96h}$ of 0.062 mg mL$^{-1}$ [31].

Behavioral changes in aquatic organisms during bioassays may be an indicator of sublethal responses induced by toxic benthic dinoflagellates [5,7,46]. All the behavioral changes (i.e., abnormal swimming activity, mobility impairment, agitation, and slow appendages beats) detected in *A. salina* nauplii exposed to *Coolia* species seem to be related to noxious effects induced by dinoflagellate toxicity. Another possibility is that the large amount of mucus secreted by *Coolia* cells at high abundances may have caused behavioral alterations in *A. salina* nauplii, particularly when exposed to *C.* cf. *canariensis* PII, which induced more sublethal effects. Extracts of *C. malayensis* strains isolated from the Pacific Ocean have induced mobility impairment in exposed *Artemia* after 8 h, as well as a severe reduction in motility after 24 h and swimming inability after 30 h [23]. Abnormal behavior in brine shrimp (e.g., imbalanced swimming and/or lack of mobility) was detected after exposure to *Coolia* lysates [36]. Abnormal swimming has been reported in tintinids after exposure to the planktonic *Alexandrium* species [47,48]. Exposure to *A. fundyense* decreased the mobility of crab larvae [49], and paralysis was reported in copepod nauplii treated with exudates of *A. tamiyavanichii* [50]. Our findings highlighted the lethal and sublethal effects of two *Coolia* species to *Artemia* nauplii and more potent effects induced by *C.* cf. *canariensis* PII, which was not tested before.

Additionally, extracellular compounds released by *C.* cf. *canariensis* PII in cell-free medium (culture filtrate) with a high cellular concentration (126,000 total cells) induced significant mortality of nauplii after 48 and 72 h of incubation. Several noxious effects of bioactive extracellular compounds (BECs) produced and released by dinoflagellate species to aquatic organisms have been described in the literature [51–54]. In a previous study, filtrates of *C. guanchica* did not induce noxious effects on *A. salina* [17]. For the first time, in the present study, significant lethality was detected on *Artemia* nauplii after 48 and 72 h of incubations with culture filtrate (cell-free medium) of *C.* cf. *canariensis* PII, suggesting the production of compounds with allelopathic potential. It is important to highlight that most of the toxic effects detected on *Artemia* nauplii seem to have been induced by their direct contact with *Coolia* cells, likely caused by the ingestion of toxic cells. The length of *C. malayensis* and *C.* cf. *canariensis* PII is, on average, 24.5 μm [19], and therefore, these cells could have been grazed by *Artemia* nauplii during the assays. In a previous study, with other three benthic dinoflagellate genera (*Prorocentrum*, *Ostreopsis*, and *Gambierdiscus*), the exposure to culture filtrates (extracellular compounds) did not affect brine shrimp survivorship, and the lethality of *A. salina* adults was directly related to their feeding on dinoflagellate cells [5].

Phenotypic plasticity and differences in toxicity can be influenced by several factors, such as nutrient limitation and different environmental (or culture) conditions, which may induce not only differences in cellular toxin content, but also in toxin profile [55]. Differences in the profile of compounds synthesized by these dinoflagellates may also cause intraspecific variations in toxicity (i.e., among strains of the same *Coolia* species) [23,36]. Toxic compounds were already detected in five *Coolia* lineages—*C. canariensis* phylogroup IV, *C. tropicalis*, *C. malayensis*, *C. palmyrensis*, and *C. santacroce* [16,21,23,27–29]. However, biological activity and toxicity to marine organisms are not directly related to the presence of known *Coolia* toxins [23]. The findings of harmful and lethal effects, particularly caused by *C.* cf. *canariensis* PII in *A. salina* nauplii, reinforce the need for further studies to determine

and identify the compounds that may be synthesized by *Coolia* species and their association with toxic effects in bioassays.

## 5. Conclusions

In the present study, for the first time, the toxicity of *C.* cf. *canariensis* PII was detected using *Artemia* bioassays. A concentration-dependent response on nauplii survival was induced by *C.* cf. *canariensis* PII exposure after 24 and 48 h. More potent effects on nauplii survival were detected after exposure to *C.* cf. *canariensis* PII with a shorter time (24 h) and concentration (330 cells mL$^{-1}$). Moreover, extracellular compounds produced by *C.* cf. *canariensis* PII and released in culture medium induced lethality to nauplii, suggesting an allelopathic potential of this strain in the environment. *Coolia malayensis* also induced lethal and sublethal effects on *Artemia* nauplii after 48 and 72 h of exposure to concentrations of 1704 and 852 cells mL$^{-1}$, respectively. Our study highlights that short-term exposure to *Coolia* cells may induce harmful effects and lethality in a marine model species.

**Author Contributions:** Conceptualization: A.M., R.A.F.N. and S.M.N.; methodology: A.M., S.M.N. and R.A.F.N.; validation: A.M. and R.A.F.N.; formal analysis: R.A.F.N.; investigation: A.M. and R.A.F.N.; writing—original draft: A.M. and R.A.F.N.; writing—review and editing: A.M., S.M.N. and R.A.F.N.; visualization: A.M., S.M.N. and R.A.F.N.; funding acquisition: S.M.N. and R.A.F.N. All authors have read and agreed to the published version of the manuscript.

**Funding:** This study was financially supported by Foundation Carlos Chagas Filho Research Support of the State of Rio de Janeiro (FAPERJ) through research grants to S.M. Nascimento (SEI-260003/002134/2021) and to R.A.F. Neves (E-26/201.283/2021), and by the Brazilian National Council for Scientific and Technological Development (CNPq) through the research grant to R.A.F. Neves (PQ2; 306212/2022-6). This study was funded in part by the Coordenação de Aperfeiçoamento de Pessoal de Nível Superior—Brasil (CAPES)—Finance Code 001—through a master's scholarship (A. Miralha).

**Institutional Review Board Statement:** Not applicable.

**Informed Consent Statement:** Not applicable.

**Data Availability Statement:** The data presented in this study are available in the manuscript.

**Acknowledgments:** The authors are grateful to Joel Campos de Paula (UNIRIO) and to anonymous reviewers for their comments and suggestions for manuscript improvement.

**Conflicts of Interest:** The authors declare no conflict of interest.

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
