# Peer review of "Coolia Species (Dinophyceae) from the Tropical South Atlantic Region: Evidence of Harmfulness of Coolia cf. canariensis Phylogroup II"

_phycology, doi:10.3390/phycology3020015_

Round 1

Reviewer 1 Report

The research was well conducted and reported.

I have only a few minor comments.

In the tables, data about swimming and motility effects should have been presented in a qualitative manner such as +, or ++ and +++.

In fig 1c the mean values were not presented to keep this figure uniform with the remaining figures.

If Artemia swims and Coolia cells sink (due to the typically slow motion of benthic species), how could Coolia cells exert toxic effects in the test tube? The authors discussed the relevance of mucous, but what about extracellular toxins? The authors avoided cell lysates, which is not so relevant in nature, but this work lacked the experimentation of culture exudates.

Author Response

Reviewer #1

The research was well conducted and reported. I have only a few minor comments. In the tables, data about swimming and motility effects should have been presented in a qualitative manner such as +, or ++ and +++.

Authors: We acknowledge Reviewer#1 suggestion. However, it was not possible to include the information as qualitative data on brine shrimp behavior in the revised version of the manuscript.  Considering that the main objective of the present study was to obtain survival data to calculate LC50 and assess dinoflagellate toxicity, detailed data on swimming or motility was not acquired during the study.

In fig 1c the mean values were not presented to keep this figure uniform with the remaining figures.

Authors: We believe that Reviewer#1 is referring to the fit curve of log-inhibition model in Figure 1 (a and b). Since survival data of brine shrimps exposed to C. cf. canariensis PII after 72 h did not fit the four-parameter logistic model, it was not generated a fit curve in Figure 1c. The authors have included this information in the caption of Figure 1 to make it clear.

If Artemia swims and Coolia cells sink (due to the typically slow motion of benthic species), how could Coolia cells exert toxic effects in the test tube?

Authors: In a previous study using other three genera of benthic dinoflagellates (Prorocentrum, Ostreopsis, and Gambierdiscus), we evaluated the grazing, behavior, and survival of adult individuals of A. salina incubated under the same experimental conditions (Neves et al., 2017). In this previous study (Neves et al., 2017), adults of A. salina fed toxic benthic dinoflagellates, since brine shrimps actively swim to catch live dinoflagellate cells, that also have movement in the water column through flagellar beats.

In the present study, it is also possible that the toxicity induced by Coolia strains to the nauplii of brine shrimps was related to the direct feeding on the dinoflagellate cells. Cells of the three Coolia strains (i.e., C. malayensis, C. tropicalis, and C. cf. canariensis PII) are, on average, 24.5 µm in length (Nascimento et al., 2019), which could be grazed by Artemia nauplii during assays. Therefore, it is possible that most of the toxic effects were induced by nauplii feeding on dinoflagellate cells. The exposure to cell-free medium (extracellular compounds), after the filtration of C. cf. canariensis PII culture with 126,000 total cells, also induced significant mortality to nauplii after 48 and 72 h of exposure. A toxic effect may also be induced by extracellular compounds, but considering the concentrations of C. cf. canariensis PII tested in this study (330-42,270 cells mL-1), extracellular compounds seem to have a fewer effect on nauplii lethality when directly exposed to Coolia cells. We have included these considerations in the Discussion section of the revised version of the manuscript.  

Nascimento, S.M., da Silva, R.A.F., Oliveira, F., Fraga, S., Salgueiro, F. Morphology and molecular phylogeny of Coolia tropicalis, Coolia malayensis and a new lineage of the Coolia canariensis species complex (Dinophyceae) isolated from Brazil. Eur. J. Phycol. 2019. 54, 484–496, doi: 10.1080/09670262.2019.1599449

Neves, R.A.F., Fernandes, T., Santos, L.N., Nascimento, S.M. Toxicity of benthic dinoflagellates on grazing, behavior and survival of the brine shrimp Artemia salina. PLoS One 2017, 12, e0175168, doi: 10.1371/journal.pone.0175168

The authors discussed the relevance of mucous, but what about extracellular toxins? The authors avoided cell lysates, which is not so relevant in nature, but this work lacked the experimentation of culture exudates.

Authors: In our previous study (Neves et al., 2017), we performed acute toxicity tests with extracellular compounds (i.e., cell-free medium) from cultures of the genera Prorocentrum, Ostreopsis, and Gambierdiscus, and no effect was found on brine shrimp survivorship.

In the present study with Coolia strains, we have performed an additional experiment with cell-free medium of cultures that induced a significant concentration-dependent effect on nauplii survival (C. cf. canariensis PII). However, in a previous submission of the manuscript, an anonymous reviewer suggested removing the results of the additional assay with cell-free medium since we have just performed with the strain that induced a concentration-dependent effect in assays with live cells. However, considering the comment provided by Reviewer #1, we have included these results (including Figure 2) and discussed them in the revised version of the manuscript.

Author Response

Reviewer 2

The authors have studied the lethal effects of 3 different Coolia species on Artemia salina. The authors have attempted to calculate LC50 of these species and they have reported the LC50 for C. cf. canariensis PII (C. PII).

Major comments

There are a few obvious issues on the data reported by the authors.The first one is about the LC50 of C. cf. canariensis PII (C. PII) at 24 hours. In table 1, the cumulative mortality of A. salina reached 50% when co-cultured with over 40000 C. PII cell per mL for 24 hours. However, the LC50 for 24 hours reported in table 2 was around 18000 cells per mL. There is a huge discrepancy between the data reported in the tables. Please clarify. Is there anything wrong in the calculations?

Authors: LC50 calculation was performed based on Assay Standard Guidelines, as presented in the chapter “Data Standardization for Results Management” (Campbell et al., 2012): “Before fitting a concentration-response curve to obtain the LC50, each well should be converted to either percent activity or percent inhibition with respect to positive and negative controls. Percent activity of all replicates from a given run for a given concentration should be averaged either by taking the mean, or preferably, taking the median.”

Campbell, R.M., Dymshitz, J., Eastwood, B.J., Emkey, R., David, P., Heerding, J.M., Johnson, D., Large, T.H., Montrose, C., Nutter, S.E., Sawyer, B.D., Sigmund, K., Smith, M., Weidner, J.R., Zink, R.W. Data Standardization for Results Management, in: Sittampalam, G.S., Grossman, A., Brimacombe, K., Arkin, M., Auld, D., Austin, C.P., Baell, J., Bejcek, B., Caaveiro, J.M.M., Xu, X. (Eds.), Assay Guidance Manual. Eli Lilly & Company and the National Center for Advancing Translational Sciences, Bethesda, 2012, pp. 1–18.

A four-parameter concentration-response curve (variable slope model) with the least squares fitting method was applied after the log-transformation of x-axis values (dinoflagellate concentrations) using the equation:

Y=Bottom + (Top-Bottom)/(1+10^((LogEC50-X)×HillSlope))

where Top and Bottom are plateaus in cells mL-1 unit,

LC50 results were validated if the fitted dose-response curves had a R2 ≥0.80 and if the percent fitting error of the LC50 (FELC50, in percentage) was <40%. The % FE LC50 was calculated by the equation (Beck et al., 2004):

% FEEC50 = SE Log EC50×Ln10×100,

where SE = standard error.”

Beck B, Chen Y-F, Dere W, Devanarayan V, Eastwood BJ, Farmen MW, Iturria SJ, Iversen PW, Kahl SD, Moore RA, Sawyer BD, Weidner J. Assay Operations for SAR Support In Sittampalam GS, Coussens NP, Brimacombe K, Grossman A, Auld D, Austin C, Baell J, Bejcek B, Chung TDY, Dahlin JL, Devanaryan V, Foley TL, Glicksman M, Hall MD, Hass JV, Inglese J, Iversen PW, Kahl SD, Kales SC, Lal-Nag M, Li Z, McGee J, McManus O, Riss T, Jr OJT, Weidner JR, Xia M, Xu X, Eds. Assay Guidance Manual. Eli Lilly & Company and the National Center for Advancing Translational Sciences, 2004, pp. 1-7.

Therefore, LC50 calculation depends on data adjustment to a curve of concentration-response using a four-parameter logistic model, not simply assessing cumulative mortality percentages. That’s the explanation for differences observed in mortality percentage data and LC50 values. However, it’s important to note that the model also calculates the 95% of confidence interval that, in the case of LC50,24h, was from 10,815 to 30,171 cells mL-1 with R² of 0.85.

The second issue is about the quality of the toxicity assay using A. salina. In table 1 and 2, without C. PII, the mortality of A. salina ranged from 1.6% to 5.8%. However, the mortality significantly increased in table 4 even without C. PII. The controls cannot be reproduced from batch to batch. It would be better to redo the toxicity assay for C. tropicalis.

Authors: All the procedures applied in Artemia bioassays considered previous studies (reviewed in Nunes et al., 2006), and used nauplii hatched from commercial cysts from the same lot and geographical origin, controlled abiotic conditions, and a standard test protocol for all the assays to reduce variability. We have included this consideration in the Material and Methods section of the revised version of the manuscript. As established for all the assays, the bioassay using C. tropicalis was composed of independent replicates and repeated twice (as described in Material and Methods section); thus, considering our data reliability, the results are not supposed to be removed.

Nunes, B.S., Carvalho, F.D., Guilhermino, L.M., Van Stappen, G. Use of the genus Artemia in ecotoxicity testing. Environ. Pollut. 2006, 144, 453–462, doi: 10.1016/j.envpol.2005.12.037

The third issue is about the mortality rates of A. salina when co-culture with C. PII at high pollution densities (27,266-54,531 cells per mL). The mortality rates reduced significantly. Please explain why?

Authors: Actually, we believe that Reviewer #2 is mentioning the strain of Coolia malayensis, instead of C. cf. canariensis PII, in which cumulative mortality was not significantly different from control at concentrations of 27,266 and 54,531 cells mL-1.

As authors have previously discussed in the first version of the manuscript, increases in A. salina survival detected at intermediary concentrations of C. cf. canariensis PII and at the higher concentrations of C. malayensis (independently of exposure time) is possibly a hormetic effect (i.e., overcompensation response to a disruption in homeostasis) (Calabrese, 1999). Hormesis is a highly frequent phenomenon independently of tested stressor, biological endpoint, and experimental model system (Calabrese et al., 2001). It is not expected that a determined intensity of a specific stressor (e.g., a toxicant concentration) could induce similar hormetic responses in different biological systems; however, a hormetic effect was previously detected in A. salina individuals exposed to the chemical compound bisphenol A (Naveira et al., 2021).

Calabrese, E.J. Evidence that hormesis represents an “overcompensation” response to a disruption in homeostasis. Ecotoxicol. Environ. Saf. 1999, 42, 135–137, doi: 10.1006/eesa.1998.1729.

Calabrese, E.J.; Baldwin, L.A. Hormesis: A generalizable and unifying hypothesis. Crit. Rev. Toxicol. 2001, 31, 353–424, doi: 10.1080/20014091111730.

Naveira, C.; Rodrigues, N.; Santos, F.S.; Santos, L.N.; Neves, R.A.F. Acute toxicity of Bisphenol A (BPA) to tropical marine and estuarine species from different trophic groups. Environ. Pollut. 2021, 268, 115911, doi: 10.1016/j.envpol.2020.115911.

Overall, the manuscript is lack of novelty. Since the lethal effects of C. malayensis and C. tropicalis as well as their extracts on Artemia species have been reported by previous studies. The only novel data is the lethal effect of C. PII on A. salina. It would be better to do more experiments to report more novel data, like investigating the effect of C. PII extract on A. salina.

Authors: Phenotypic plasticity and differences in toxicity of dinoflagellates can be influenced by several factors (e.g., different environmental conditions) that may undergo not only differences in toxin content but also in the toxin profile produced by those cells. Moreover, intraspecific variations in toxicity (i.e., among strains of the same Coolia species) have been related to differences in the profile of compounds synthesized by these dinoflagellates (Leung et al., 2017; Phua et al., 2021). Therefore, independently of previous tests using a Coolia species, the toxicity assessment of three strains from different Coolia species isolated from the tropical region (including a strain that solely comprises a phylogroup) is a novel result, especially considering bioassays applying environmentally relevant dinoflagellate concentrations.

Leung, P.T.Y., Yan, M., Yiu, S.K.F., Lam, V.T.T., Ip, J.C.H., Au, M.W.Y., Chen, C.-Y., Wai, T.-C., Lam, P.K.S. Molecular phylogeny and toxicity of harmful benthic dinoflagellates Coolia (Ostreopsidaceae, Dinophyceae) in a sub-tropical marine ecosystem: The first record from Hong Kong. Mar. Pollut. Bull. 2017, 124, 878–889, doi: 10.1016/j.marpolbul.2017.01.017

Phua, Y.H., Roy, M.C., Lemer, S., Husnik, F., Wakeman, K.C. Diversity and toxicity of Pacific strains of the benthic dinoflagellate Coolia (Dinophyceae), with a look at the Coolia canariensis species complex. Harmful Algae 2021, 109, 102120,  doi:10.1016/j.hal.2021.102120

Round 2

Reviewer 2 Report

2nd round review

Manuscript ID: phycology-2303557

Manuscript title: Coolia species (Dinophyceae) from the tropical South Atlantic region: evidence of harmfulness of Coolia cf. canariensis phylogroup II

Reviewer recommendation: Major revision

Following up the 2nd issue discussed in the 1st round of review, my concern is about the high mortality rate of Artemia even without C. tropicalis reported in table 4. High mortality rates of Artemia in the controls basically meant at least some of the Artemia were not health, so they dead without stresses. Moreover, the mortality rates of the controls were high in both independent duplicated trials. Then, it is suspected that poor health conditions of Artemia may be a significant factor affecting the mortality rates in other treatment groups co-cultured with different concentrations of C. tropicalis. Please try to redo that set experiments.

Please explain the hormesis observed when Artemia were treated with different concentrations of C. malayensis in the discussion.    

Author Response

Reviewer #2: Following up the 2nd issue discussed in the 1st round of review, my concern is about the high mortality rate of Artemia even without C. tropicalis reported in table 4. High mortality rates of Artemia in the controls basically meant at least some of the Artemia were not health, so they dead without stresses. Moreover, the mortality rates of the controls were high in both independent duplicated trials. Then, it is suspected that poor health conditions of Artemia may be a significant factor affecting the mortality rates in other treatment groups co-cultured with different concentrations of C. tropicalis. Please try to redo that set experiments.

Authors: Unfortunately, it is not simply to redo the set of experiments with C. tropicalis since we must increase algal biomass of the strain in culture, which take months to be ready for assays. However, if there is a suspicious that anything went wrong with C. tropicalis set experiments, authors prefer to remove this entire set from the manuscript. We believe that removing this bioassay, we will not lose the most important finding of the present study that was the toxicity induced by C. cf. canariensis PII. Therefore, we have removed from the second version of the revised manuscript all the information concerning C. tropicalis assay.

 Reviewer #2: Please explain the hormesis observed when Artemia were treated with different concentrations of C. malayensis in the discussion.    

Authors: We included the effect observed in C. malayensis treatment in the revised version of the Discussion section. 

Round 3

Reviewer 2 Report

The revised manuscript is okay for publication